# SECURE BYZANTINE-ROBUST MACHINE LEARNING

## ABSTRACT

Increasingly machine learning systems are being deployed to edge servers and devices (e.g. mobile phones) and trained in a collaborative manner. Such distributed/federated/decentralized training raises a number of concerns about the robustness, privacy, and security of the procedure. While extensive work has been done in tackling with robustness, privacy, or security individually, their combination has rarely been studied. In this paper, we propose a secure two-server protocol that offers both input privacy and Byzantine-robustness. In addition, this protocol is communication-efficient, fault-tolerant and enjoys local differential privacy.

## 1 INTRODUCTION

Recent years have witnessed fast growth of successful machine learning applications based on data collected from decentralized user devices. Unfortunately, however, currently most of the important machine learning models on a societal level do not have their utility, control, and privacy aligned with the data ownership of the participants. This issue can be partially attributed to a fundamental conflict between the two leading paradigms of traditional centralized training of models on one hand, and decentralized/collaborative training schemes on the other hand. While centralized training violates the privacy rights of participating users, existing alternative training schemes are typically not robust. Malicious participants can sabotage the training system by feeding it wrong data intentionally, known as *data poisoning*. In this paper, we tackle this problem and propose a novel distributed training framework which offers both *privacy* and *robustness*.

When applied to datasets containing personal data, the use of privacy-preserving techniques is currently required under regulations such as the *General Data Protection Regulation* (GDPR) or *Health Insurance Portability and Accountability Act* (HIPAA). The idea of training models on decentralized datasets and incrementally aggregating model updates via a central server motivates the federated learning paradigm (McMahan et al., 2016). However, the averaging in federated learning, when viewed as a *multi-party computation (MPC)*, does not preserve the *input privacy* because the server observes the models directly. The *input privacy* requires each party learns nothing more than the output of computation which in this paradigm means the aggregated model updates. To solve this problem, *secure* aggregation rules as proposed in (Bonawitz et al., 2017) achieve guaranteed input privacy. Such secure aggregation rules have found wider industry adoption recently e.g. by Google on Android phones (Bonawitz et al., 2019; Ramage & Mazzocchi, 2020) where input privacy guarantees can offer e.g. efficiency and exactness benefits compared to differential privacy (both can also be combined).

The concept of Byzantine robustness has received considerable attention in the past few years for practical applications, as a way to make the training process robust to malicious actors. A Byzantine participant or worker can behave arbitrarily malicious, e.g. sending arbitrary updates to the server. This poses great challenge to the most widely used aggregation rules, e.g. simple average, since a single Byzantine worker can compromise the results of aggregation. A number of Byzantine-robust aggregation rules have been proposed recently (Blanchard et al., 2017; Muñoz-González et al., 2017; Alistarh et al., 2018; Mhamdi et al., 2018; Yin et al., 2018; Muñoz-González et al., 2019) and can be used as a building block for our proposed technique.

Achieving both input privacy and Byzantine robustness however remained elusive so far, with Bagdasaryan et al. (2020) stating that robust rules "*...are incompatible with secure aggregation*". We here prove that this is not the case. Closest to our approach is (Pillutla et al., 2019) which tolerates data poisoning but does not offer Byzantine robustness. Prio (Corrigan-Gibbs & Boneh, 2017) is a private and robust aggregation system relying on secret-shared non-interactive proofs (SNIP). While

their setting is similar to ours, the robustness they offer is limited to check the range of the input. Besides, the encoding for SNIP has to be affine-aggregable and is expensive for clients to compute.

In this paper, we propose a secure aggregation framework with the help of two non-colluding honest-but-curious servers. This framework also tolerates server-worker collusion. In addition, we combine robustness and privacy at the cost of leaking only worker similarity information which is marginal for high-dimensional neural networks. Note that our focus is not to develop new defenses against state-of-the-art attacks, e.g. (Baruch et al., 2019; Xie et al., 2019b). Instead, we focus on making *arbitary* current and future distance-based robust aggregation rules (e.g. Krum by Mhamdi et al. (2018), RFA by Pillutla et al. (2019)) compatible with secure aggregation.

**Main contributions.** We propose a novel distributed training framework which is

- **Privacy-preserving:** our method keeps the input data of each user secure against any other user, and against our honest-but-curious servers.
- **Byzantine robust:** our method offers Byzantine robustness and allows to incorporate existing robust aggregation rules, e.g. (Blanchard et al., 2017; Alistarh et al., 2018). The results are exact, i.e. identical to the non-private robust methods.
- **Fault tolerant and easy to use:** our method natively supports workers dropping out or newly joining the training process. It is also easy to implement and to understand for users.
- **Efficient and scalable:** the computation and communication overhead of our method is negligible (less than a factor of 2) compared to non-private methods. Scalability in terms of cost including setup and communication is linear in the number of workers.

## 2 PROBLEM SETUP, PRIVACY, AND ROBUSTNESS

We consider the distributed setup of $n$ user devices, which we call workers, with the help of two additional servers. Each worker $i$ has its own private part of the training dataset. The workers want to collaboratively train a public model benefitting from the joint training data of all participants.

In every training step, each worker computes its own private model update (e.g. a gradient based on its own data) denoted by the vector $x_i$. The aggregation protocol aims to compute the sum $z = \sum_{i=1}^{n} x_i$ (or a robust version of this aggregation), which is then used to update a public model. While the result $z$ is public in all cases, the protocol must keep each $x_i$ private from any adversary or other workers.

**Security model.** We consider honest-but-curious servers which do not collude with each other but may collude with malicious workers. An honest-but-curious server follows the protocol but may try to inspect all messages. We also assume that all communication channels are secure. We guarantee the strong notion of *input privacy*, which means the servers and workers know nothing more about each other than what can be inferred from the public output of the aggregation $z$.

**Byzantine robustness model.** We allow the standard Byzantine worker model which assumes that workers can send arbitrary adversarial messages trying to compromise the process. We assume that a fraction of up to $\alpha$ ($< 0.5$) of the workers is Byzantine, i.e. are *malicious* and not follow the protocol.

**Additive secret sharing.** Secret sharing is a way to split any secret into multiple parts such that no part leaks the secret. Formally, suppose a scalar $a$ is a *secret* and the secret holder shares it with $k$ parties through *secret-shared values* $\langle a \rangle$. In this paper, we only consider additive secret-sharing where $\langle a \rangle$ is a notation for the set $\{a_i\}_{i=1}^{k}$ which satisfy $a = \sum_{p=1}^{k} a_p$, with $a_p$ held by party $p$. Crucially, it must not be possible to reconstruct $a$ from any $a_p$. For vectors like $x$, their secret-shared values $\langle x \rangle$ are simply component-wise scalar secret-shared values.

**Two-server setting.** We assume there are two non-colluding servers: model server (S1) and worker server (S2). S1 holds the output of each aggregation and thus also the machine learning model which is public to all workers. S2 holds intermediate values to perform Byzantine aggregation. Another key assumption is that the servers have no incentive to collude with workers, perhaps enforced via a potential huge penalty if exposed. It is realistic to assume that the communication link between the two servers S1 and S2 is faster than the individual links to the workers. To perform robust aggregation, the servers will need access to a sufficient number of *Beaver's triples*. These are data-independent values required to implement secure multiplication in MPC on both servers, and can be precomputed beforehand. For completeness, the classic algorithm for multiplication is given in in Appendix B.1.

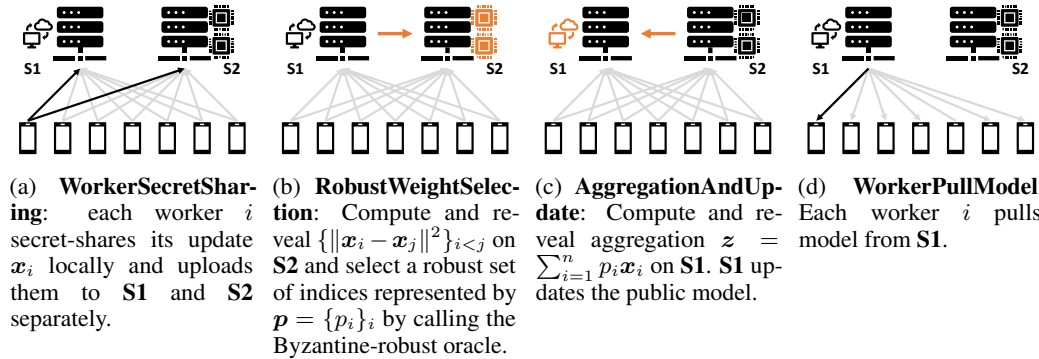

(a) **WorkerSecretSharing**: each worker $i$ secret-shares its update $\boldsymbol{x}_i$ locally and uploads them to **S1** and **S2** separately.

(b) **RobustWeightSelection**: Compute and reveal $\{\|\boldsymbol{x}_i - \boldsymbol{x}_j\|^2\}_{i<j}$ on **S2** and select a robust set of indices represented by $\boldsymbol{p} = \{p_i\}_i$ by calling the Byzantine-robust oracle.

(c) **AggregationAndUpdate**: Compute and reveal aggregation $\boldsymbol{z} = \sum_{i=1}^{n} p_i \boldsymbol{x}_i$ on **S1**. **S1** updates the public model.

(d) **WorkerPullModel**: Each worker $i$ pulls model from **S1**.

Figure 1: Illustration of Algorithm 2. The orange components on servers represent the computation-intensive operations at low communication cost between servers.

**Byzantine-robust aggregation oracles.** Most of existing robust aggregation algorithms rely on distance measures to identity potential adversarial behavior (Blanchard et al., 2017; Yin et al., 2018; Mhamdi et al., 2018; Li et al., 2019; Ghosh et al., 2019). All such distance-based aggregation rules can be directly incorporated into our proposed scheme, making them secure. While many aforementioned papers assume that the workers have i.i.d datasets, our protocol is oblivious to the distribution of the data across the workers. In particular, our protocol also works with schemes such as (Li et al., 2019; Ghosh et al., 2019; He et al., 2020) designed for non-iid data.

## 3 SECURE AGGREGATION PROTOCOL: TWO-SERVER MODEL

Each worker first splits its private vector $\boldsymbol{x}_i$ into two additive secret shares, and transmits those to each corresponding server, ensuring that neither server can reconstruct the original vector on its own. The two servers then execute our secure aggregation protocol. On the level of servers, the protocol is a two-party computation (2PC). In the case of non-robust aggregation, servers simply add all shares (we present this case in detail in Algorithm 1). In the robust case which is of our main interest here, the two servers exactly emulate an existing Byzantine robust aggregation rule, at the cost of revealing only distances of worker gradients on the server (the robust algorithm is presented in Algorithm 2). Finally, the resulting aggregated output vector $\boldsymbol{z}$ is sent back to all workers and applied as the update to the public machine learning model.

### 3.1 NON-ROBUST SECURE AGGREGATION

In each round, Algorithm 1 consists of two stages:

- **WorkerSecretSharing** (Figure 1a): each worker $i$ randomly splits its private input $\boldsymbol{x}_i$ into two additive secret shares $\boldsymbol{x}_i = \boldsymbol{x}_i^{(1)} + \boldsymbol{x}_i^{(2)}$. This can be done e.g. by sampling a large noise value $\xi_i$ and then using $(\boldsymbol{x}_i \pm \xi_i)/2$ as the shares. Worker $i$ sends $\boldsymbol{x}_i^{(1)}$ to **S1** and $\boldsymbol{x}_i^{(2)}$ to **S2**. We write $\langle \boldsymbol{x}_i \rangle$ for the two secret-shared values distributed over the two servers.
- **AggregationAndUpdate** (Figure 1c): Given binary weights $\{p_i\}_{i=1}^{n}$, each server locally computes $\langle \sum_{i=1}^{n} p_i \boldsymbol{x}_i \rangle$. Then **S2** sends its share $\langle \sum_{i=1}^{n} p_i \boldsymbol{x}_i \rangle^{(2)}$ to **S1** so that **S1** can then compute $\boldsymbol{z} = \sum_{i=1}^{n} p_i \boldsymbol{x}_i$. **S1** updates the public model with $\boldsymbol{z}$.

Our secure aggregation protocol is extremely simple, and as we will discuss later, has very low communication overhead, does not require heavy cryptographic primitives, gives strong input privacy and is compatible with differential privacy, and is robust to worker dropouts and failures. We believe this makes our protocol especially attractive for federated learning applications.

We now argue about correctness and privacy. It is clear that the output $\boldsymbol{z}$ of the above protocol satisfies $\boldsymbol{z} = \sum_{i=1}^{n} p_i \boldsymbol{x}_i$, ensuring that all workers compute the right update. Now we argue about the privacy guarantees. We track the values stored by each of the servers and workers:

- **S1**: Its own secret shares $\{\boldsymbol{x}_i^{(1)}\}_{i=1}^{n}$ and the sum of the other shares $\langle \sum_{i=1}^{n} p_i \boldsymbol{x}_i \rangle^{(2)}$.
- **S2**: Its own secret shares $\{\boldsymbol{x}_i^{(2)}\}_{i=1}^{n}$.
- Worker $i$: $\boldsymbol{x}_i$ and $\boldsymbol{z} = \sum_{i=1}^{n} p_i \boldsymbol{x}_i$.

Clearly, the workers have no information other than the aggregate $\boldsymbol{z}$ and their own data. **S2** only has the secret share which on their own leak no information about any data. Hence surprisingly, **S2** does not learn anything in this process. **S1** has its own secret share and also the sum of the other shares. If $n = 1$, then $\boldsymbol{z} = \boldsymbol{x}_i$ and hence **S1** is allowed to learn everything. If $n > 1$, then **S1** cannot recover information about any individual secret share $\boldsymbol{x}_i^{(2)}$ from the sum. Thus, **S1** learns $\boldsymbol{z}$ and nothing else.

## 3.2 ROBUST SECURE AGGREGATION

We now describe how Algorithm 2 replaces the simple aggregation with any distance-based robust aggregation rule **Aggr**, e.g. Multi-Krum (Blanchard et al., 2017). The key idea is to use two-party MPC to securely compute multiplication.

- **WorkerSecretSharing** (Figure 1a): As before, each worker $i$ secret shares $\langle \boldsymbol{x}_i \rangle$ distributed over the two servers **S1** and **S2**.
- **RobustWeightSelection** (Figure 1b): After collecting all secret-shared values $\{\langle \boldsymbol{x}_i \rangle\}_i$, the servers compute pairwise difference $\{\langle \boldsymbol{x}_i - \boldsymbol{x}_j \rangle\}_{i<j}$ locally. **S2** then reveals—to itself exclusively—in plain text all of the pairwise Euclidean distances between workers $\{\|\boldsymbol{x}_i - \boldsymbol{x}_j\|^2\}_{i<j}$ with the help of precomputed Beaver's triples and Algorithm 3. The distances are kept private from **S1** and workers. **S2** then feeds these distances to the distance-based robust aggregation rule **Aggr**, returning (on **S2**) a binary weight vector $\boldsymbol{p} = \{p_i\}_{i=1}^n \in \{0,1\}^n$, representing the indices of the robust subset selected by **Aggr**.
- **AggregationAndUpdate** (Figure 1c): Given weight vector $\boldsymbol{p}$ from previous step, we would like S1 to compute $\sum_{i=1}^n p_i \boldsymbol{x}_i$. To do so, **S2** secret shares with **S1** the values of $\{\langle p_i \rangle\}$ instead of sending in plain-text since they may be private. Then, **S1** reveals to itself, but not to **S2**, in plain text the value of $\boldsymbol{z} = \sum_{i=1}^n p_i \boldsymbol{x}_i$ using secret-shared multiplication and updates the public model.
- **WorkerPullModel** (Figure 1d): Workers pull the latest public model on **S1** and update it locally.

The key difference between the robust and the non-robust aggregation scheme is the weight selection phase where **S2** computes all pairwise distances and uses this to run a robust-aggregation rule in a black-box manner. **S2** computes these distances i) without leaking any information to **S1**, and ii) without itself learning anything other than the pair-wise distances (and in particular none of the actual values of $\boldsymbol{x}_i$). To perform such a computation, **S1** and **S2** use precomputed *Beaver's triplets* (Algorithm 3 in the Appendix), which can be made available in a scalable way (Smart & Tanguy, 2019).

## 3.3 SALIENT FEATURES

Overall, our protocols are very resource-light and straightforward from the perspective of the workers. Further, since we use Byzantine-robust aggregation, our protocols are provably fault-tolerant even if a large fraction of workers misbehave. This further lowers the requirements of a worker. We eleborate the features as follows.

**Communication overhead.** In applications, individual uplink speed from worker and servers is typically the main bottleneck, as it is typically much slower than downlink, and the bandwidth between servers can be very large. For our protocols, the time spent on the uplink is within a factor of 2 of the non-secure variants. Besides, our protocol only requires one round of communication, which is an advantage over interactive proofs.

**Fault tolerance.** The workers in Algorithm 1 and Algorithm 2 are completely stateless across multiple rounds and there is no *offline* phase required. This means that workers can start participating in the protocols simply by pulling the latest public model. Further, our protocols are unaffected if some workers drop out in the middle of a round. Unlike in (Bonawitz et al., 2017), there is no entanglement between workers and we don't have unbounded recovery issues.

**Compatibility with local differential privacy.** One byproduct of our protocol can be used to convert differentially private mechanisms, such as (Abadi et al., 2016) which only guarantees the privacy of the aggregated model, into the stronger *locally* differentially private mechanisms which guarantee user-level privacy.

---

**Algorithm 1** Two-Server Secure Aggregation (Non-robust variant)

---

**Setup**: $n$ workers (non-Byzantine) with private vectors $\boldsymbol{x}_i$. Two non-colluding servers **S1** and **S2**.
**Workers**:  (**WorkerSecretSharing**)
 1. split private $\boldsymbol{x}_i$ into additive secret shares $\langle \boldsymbol{x}_i \rangle = \{\boldsymbol{x}_i^{(1)}, \boldsymbol{x}_i^{(2)}\}$ (such that $\boldsymbol{x}_i = \boldsymbol{x}_i^{(1)} + \boldsymbol{x}_i^{(2)}$)
 2. send $\boldsymbol{x}_i^{(1)}$ to **S1** and $\boldsymbol{x}_i^{(2)}$ to **S2**
**Servers**:
 1. $\forall\, i$, **S1** collects $\boldsymbol{x}_i^{(1)}$ and **S2** collects $\boldsymbol{x}_i^{(2)}$
 2. (**AggregationAndUpdate**):
    (a) On **S1** and **S2**, compute $\langle \sum_{i=1}^{n} \boldsymbol{x}_i \rangle$ locally
    (b) **S2** sends its share of $\langle \sum_{i=1}^{n} \boldsymbol{x}_i \rangle$ to **S1**
    (c) **S1** reveals $\boldsymbol{z} = \sum_{i=1}^{n} \boldsymbol{x}_i$ to everyone

---

**Algorithm 2** Two-Server Secure Robust Aggregation (Distance-Based)

---

**Setup**: $n$ workers, $\alpha n$ of which are Byzantine. Two non-colluding servers **S1** and **S2**.
**Workers**:  (**WorkerSecretSharing**)
 1. split private $\boldsymbol{x}_i$ into additive secret shares $\langle \boldsymbol{x}_i \rangle = \{\boldsymbol{x}_i^{(1)}, \boldsymbol{x}_i^{(2)}\}$ (such that $\boldsymbol{x}_i = \boldsymbol{x}_i^{(1)} + \boldsymbol{x}_i^{(2)}$)
 2. send $\boldsymbol{x}_i^{(1)}$ to **S1** and $\boldsymbol{x}_i^{(2)}$ to **S2**
**Servers**:
 1. $\forall\, i$, **S1** collects gradient $\boldsymbol{x}_i^{(1)}$ and **S2** collects $\boldsymbol{x}_i^{(2)}$
 2. (**RobustWeightSelection**):
    (a) For each pair $(\boldsymbol{x}_i,\ \boldsymbol{x}_j)$ compute their Euclidean distance $(i < j)$:
       • On **S1** and **S2**, compute $\langle \boldsymbol{x}_i - \boldsymbol{x}_j \rangle = \langle \boldsymbol{x}_i \rangle - \langle \boldsymbol{x}_j \rangle$ locally
       • Use precomputed Beaver's triples (see Algorithm 3) to compute the distance $\|\boldsymbol{x}_i - \boldsymbol{x}_j\|^2$
    (b) **S2** perform robust aggregation rule $\boldsymbol{p} = $**Aggr**$(\{\|\boldsymbol{x}_i - \boldsymbol{x}_j\|^2\}_{i<j})$
    (c) **S2** secret-shares $\langle \boldsymbol{p} \rangle$ with **S1**
 3. (**AggregationAndUpdate**):
    (a) On **S1** and **S2**, use MPC multiplication to compute $\langle \sum_{i=1}^{n} p_i \boldsymbol{x}_i \rangle$ locally
    (b) **S2** sends its share of $\langle \sum_{i=1}^{n} p_i \boldsymbol{x}_i \rangle^{(2)}$ to **S1**
    (c) **S1** reveals $\boldsymbol{z} = \sum_{i=1}^{n} p_i \boldsymbol{x}_i$ to all workers.
**Workers**:
 1. (**WorkerPullModel**): Collect $\boldsymbol{z}$ and update model locally

---

**Other Byzantine-robust oracles.** We can also use some robust-aggregation rules which are not based on pair-wise distances such as Byzantine SGD (Alistarh et al., 2018). Since the basic structures are very similar to Algorithm 2, we put Algorithm 8 in the appendix.

**Security.** The security of Algorithm 1 is straightforward as we previously discussed. The security of Algorithm 2 again relies on the separation of information between **S1** and **S2** with neither the workers nor **S1** learning anything other than the aggregate $\boldsymbol{z}$. We will next formally prove that this is true even in the presence of malicious workers.

**Remark 1.** *Our proposed scheme leverages classic 2-party secret-sharing for addition and multiplication. These building blocks however are originally proposed for integers and quantized values, not real values. For floating point operations as used in machine learning, one can use the secure counterparts (Aliasgari et al., 2013) of the two operations. This is facilitated by deep learning training being robust to limited precision training (Gupta et al., 2015) and additional noise (Neelakantan et al., 2016), with current models routinely trained in 16 bit precision. In contrast to (Bonawitz et al., 2017) which relies on advanced cryptographic primitives such as Diffie-Hellman's key agreement which must remain exact and discrete, our protocols only use much simpler secure arithmetic operations—only addition and multiplication—which are tolerant to rounding errors. For the privacy implications of secret sharing when using floating point, which go beyond the scope of our work, we refer the reader to the information theoretic analysis of Aliasgari et al. (2013).*

# 4 THEORETICAL GUARANTEES

## 4.1 EXACTNESS

In the following lemma we show that Algorithm 2 gives the exact same result as non-privacy-preserving version.

**Lemma 2** (Exactness of Algorithm 2). *The resulting $\boldsymbol{z}$ in Algorithm 2 is identical to the output of the non-privacy-preserving version of the used robust aggregation rule.*

*Proof.* After secret-sharing $\boldsymbol{x}_i$ to $\langle \boldsymbol{x}_i \rangle$ to two servers, Algorithm 2 performs local differences $\{\langle \boldsymbol{x}_i - \boldsymbol{x}_j \rangle\}_{i<j}$. Using shared-values multiplication via Beaver's triple, **S2** obtains the list of true Euclidean distances $\{\|\boldsymbol{x}_i - \boldsymbol{x}_j\|^2\}_{i<j}$. The result is fed to a *distance-based* robust aggregation rule oracle, all solely on **S2**. Therefore, the resulting indices $\{p_i\}_i$ as used in $\boldsymbol{z} := \Sigma_{i=1}^n p_i \boldsymbol{x}_i$ are identical to the aggregation of non-privacy-preserving robust aggregation. $\square$

With the exactness of the protocol established, we next focus on the privacy guarantee.

## 4.2 PRIVACY

We prove probabilistic (information-theoretic) notion of privacy which gives the strongest guarantee possible. Formally, we will show that the distribution of the secret does not change even after being conditioned on all observations made by all participants, i.e. each worker $i$, **S1** and **S2**. This implies that the observations carry absolutely no information about the secret. Our results rely on the existence of simple additive secret-sharing protocols as discussed in the Appendix.

Each worker $i$ only receives the final aggregated $\boldsymbol{z}$ at the end of the protocol and is not involved in any other manner. Hence no information can be leaked to them. We will now examine **S1**. The proofs below rely on Beaver's triples which we summarize in the following lemma.

**Lemma 3** (Beaver's triples). *Suppose we secret share $\langle x \rangle$ and $\langle y \rangle$ between **S1** and **S2** and want to compute $xy$ on **S2**. There exists a protocol which enables such computation which uses precomputed shares $BV = (\langle a \rangle, \langle b \rangle, \langle c \rangle)$ such that **S1** does not learn anything and **S2** only learns $xy$.*

Due to the page limit, we put the details about Beaver's triples, multiplying secret shares, as well as the proofs for the next two theorems to the Appendix.

**Theorem I** (Privacy for **S1**). *Let $\boldsymbol{z} = \sum_{i=1}^n p_i \boldsymbol{x}_i$ where $\{p_i\}_{i=1}^n$ is the output of byzantine oracle or a vector of 1s (non-private). Let $BV_{ij} = \langle \boldsymbol{a}_{ij}, \boldsymbol{b}_{ij}, \boldsymbol{c}_{ij} \rangle$ and $BVp_i = \langle \boldsymbol{a}_i^p, \boldsymbol{b}_i^p, \boldsymbol{c}_i^p \rangle$ be the Beaver's triple used in the multiplications. Let $\langle \cdot \rangle^{(1)}$ be the share of the secret-shared values $\langle \cdot \rangle$ on **S1**. Then for all workers $i$*

$$\mathbb{P}(\boldsymbol{x}_i = x_i \mid \{\langle \boldsymbol{x}_i \rangle^{(1)}, \langle p_i \rangle^{(1)}\}_{i=1}^n, \{BV_{i,j}^{(1)}, \boldsymbol{x}_i - \boldsymbol{x}_j - \boldsymbol{a}_{ij}, \boldsymbol{x}_i - \boldsymbol{x}_j - \boldsymbol{b}_{ij}\}_{i<j},$$

$$\{\langle \|\boldsymbol{x}_i - \boldsymbol{x}_j\|^2 \rangle^{(1)}\}_{i<j}, \{BVp_i^{(1)}, p_i - \boldsymbol{a}_i^p, p_i - \boldsymbol{b}_i^p\}_{i=1}^n, \boldsymbol{z}) = \mathbb{P}(\boldsymbol{x}_i = x_i | \boldsymbol{z})$$

*Note that the conditioned values are what **S1** observes throughout the algorithm. $\{BV_{ij}^{(1)}, \boldsymbol{x}_i - \boldsymbol{x}_j - \boldsymbol{a}_{ij}, \boldsymbol{x}_i - \boldsymbol{x}_j - \boldsymbol{b}_{ij}\}_{i<j}$ and $\{BVp_i^{(1)}, p_i - \boldsymbol{a}_i^p, p_i - \boldsymbol{b}_i^p\}_{i=1}^n$ are intermediate values during shared values multiplication.*

For **S2**, the theorem to prove is a bit different because in this case **S2** doesn't know the output of aggregation $\boldsymbol{z}$. In fact, this is more similar to an independent system which knows little about the underlying tasks, model weights, etc. We show that while **S2** has observed many intermediate values, it can only learn no more than what can be inferred from model distances.

**Theorem II** (Privacy for **S2**). *Let $\{p_i\}_{i=1}^n$ is the output of byzantine oracle or a vector of 1s (non-private). Let $BV_{ij} = \langle \boldsymbol{a}_{ij}, \boldsymbol{b}_{ij}, \boldsymbol{c}_{ij} \rangle$ and $BVp_i = \langle \boldsymbol{a}_i^p, \boldsymbol{b}_i^p, \boldsymbol{c}_i^p \rangle$ be the Beaver's triple used in the multiplications. Let $\langle \cdot \rangle^{(2)}$ be the share of the secret-shared values $\langle \cdot \rangle$ on **S2**. Then for all workers $i$*

$$\mathbb{P}(\boldsymbol{x}_i = x_i \mid \{\langle \boldsymbol{x}_i \rangle^{(2)}, \langle p_i \rangle^{(2)}, p_i\}_{i=1}^n, \{BV_{i,j}^{(2)}, \boldsymbol{x}_i - \boldsymbol{x}_j - \boldsymbol{a}_{ij}, \boldsymbol{x}_i - \boldsymbol{x}_j - \boldsymbol{b}_{ij}\}_{i<j},$$

$$\{\langle \|\boldsymbol{x}_i - \boldsymbol{x}_j\|^2 \rangle^{(2)}, \|\boldsymbol{x}_i - \boldsymbol{x}_j\|^2\}_{i<j}, \{BVp_i^{(2)}, p_i - \boldsymbol{a}_i^p, p_i - \boldsymbol{b}_i^p\}_{i=1}^n) \qquad (1)$$

$$= \mathbb{P}(\boldsymbol{x}_i = x_i \mid \{\|\boldsymbol{x}_i - \boldsymbol{x}_j\|^2\}_{i<j})$$

*Note that the conditioned values are what **S2** observed throughout the algorithm. $\{BV_{ij}^{(2)}, \boldsymbol{x}_i - \boldsymbol{x}_j - \boldsymbol{a}_{ij}, \boldsymbol{x}_i - \boldsymbol{x}_j - \boldsymbol{b}_{ij}\}_{i<j}$ and $\{BVp_i^{(2)}, p_i - \boldsymbol{a}_i^p, p_i - \boldsymbol{b}_i^p\}_{i=1}^n$ are intermediate values during shared values multiplication.*

The model distances indeed only leaks similarity among the workers. Such similarity, however, does not tell **S2** information about the parameters; in (Mhamdi et al., 2018) the *leeway attack* attacks distance based-rules because they don't distinguish two gradients with evenly distributed noise and two different gradients very different in one parameter. This means the leaked information has low impact to the privacy.

It is also worth noting that curious workers can only inspect others' values by learning from the public model/update. This is because in our scheme, workers don't interact directly and there is only one round of communication between servers and workers. So the only message a worker receives is the public model update.

### 4.3 COMBINING WITH DIFFERENTIAL PRIVACY

While input privacy is our main goal, our approach is naturally compatible with other orthogonal notions of privacy. Global differential privacy (DP) (Shokri & Shmatikov, 2015; Abadi et al., 2016; Chase et al., 2017) is mainly concerned about the privacy of the *aggregated* model, and whether it leaks information about the training data. On the other hand, local differential privacy (LDP) (Evfimievski et al., 2003; Kasiviswanathan et al., 2011) is stronger notions which is also concerned with the training process itself. It requires that every communication transmitted by the worker does not leak information about their data. In general, it is hard to learn deep learning models satisfying LDP using iterate perturbation (which is the standard mechanism for DP) (Bonawitz et al., 2017).

Our non-robust protocol *is naturally compatible* with local differential privacy. Consider the usual iterative optimization algorithm which in each round $t$ performs

$$\boldsymbol{w}_t \leftarrow \boldsymbol{w}_{t-1} - \eta(\boldsymbol{x}_t + \nu_t), \text{ where } \boldsymbol{x}_t = \frac{1}{n}\sum_{i=1}^n \boldsymbol{x}_{t,i}. \tag{2}$$

Here $\boldsymbol{x}_t$ is the aggregate update, $\boldsymbol{w}_t$ is the model parameters, and $\nu_t$ is the noise added for DP (Abadi et al., 2016).

**Theorem III** (from DP to LDP). *Suppose that the noise $\nu_t$ in (2) is sufficient to ensure that the set of model parameters $\{\boldsymbol{w}_t\}_{t\in[T]}$ satisfy $(\varepsilon, \delta)$-DP for $\varepsilon \geq 1$. Then, running (2) with using Alg. 1 to compute $(\boldsymbol{x}_t + \eta_t)$ by securely aggregating $\{\boldsymbol{x}_{1,t} + n\eta_t, \boldsymbol{x}_{2,t}, \ldots, \boldsymbol{x}_{n,t}\}$ satisfies $(\varepsilon, \delta)$-LDP.*

Unlike existing approaches, we do not face a tension between differential privacy which relies on real-valued vectors and cryptographic tools which operate solely on discrete/quantized objects. This is because our protocols do not rely on cryptographic primitives like Diffie-Hellman key agreement, in contrast to e.g. (Bonawitz et al., 2017). In particular, the vectors $\boldsymbol{x}_i$ can be full-precision (real-valued) at the cost of adding marginal rounding error which can be tolerated by robust aggregation rule and stochastic gradient descent algorithms. Thus, our secure aggregation protocol can be integrated with a mechanism which has global DP properties e.g. (Abadi et al., 2016), and prove *local* DP guarantees for the resulting mechanism.

## 5 EMPIRICAL ANALYSIS OF OVERHEAD

We present an illustrative simulation on a local machine (i7-8565U) to demonstrate the overhead of our scheme. We use PyTorch with MPI to train a neural network of 1.2 million parameters on the MNIST dataset. We compare the following three settings: simple aggregation with 1 server, secure aggregation with 2 servers, robust secure aggregation with 2 servers (with Krum (Blanchard et al., 2017)). The number of workers is always 5.

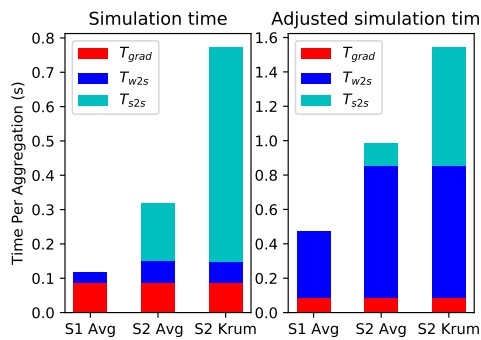

Figure 2 shows the time spent on all parts of training for one aggregation step. $T_{grad}$ is the

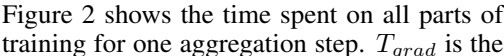

Figure 2: Left: Actual time spent; Right: Time adjusted for network bandwidth.

time spent on batch gradient computation; $T_{w2s}$ refers to the time spend on uploading and downloading gradients; $T_{s2s}$ is the time spend on communication between servers. Note that the server-to-server communication could be further reduced by employing more efficient aggregationn rules. Since the simulation is run on a local machine, time spent on communication is underestimated. In the right hand side figure, we adjusts time by assuming the worker-to-server link has 100Mbps bandwidth and 1Gbps respectively for the server-to-server link. Even in this scenario, we can see that the overhead from private aggregation is small. Furthermore, the additional overhead by the robustness module is moderate comparing to the standard training, even for realistic deep-learning settings. For comparison, a zero-knowledge-proof-based approach need to spend 0.03 seconds to encode a submission of 100 integers (Corrigan-Gibbs & Boneh, 2017).

## 6 LITERATURE REVIEW

**Secure Aggregation.** In the standard distributed setting with 1 server, Bonawitz et al. (2017) proposes a secure aggregation rule which is also fault tolerant. They generate a shared secret key for each pair of users. The secret keys are used to construct masks to the input gradients so that masks cancel each other after aggregation. To achieve fault tolerance, they employ Shamir's secret sharing. To deal with active adversaries, they use a public key infrastructure (PKI) as well as a second mask applied to the input. A followup work (Mandal et al., 2018) minimizes the pairwise communication by outsourcing the key generation to two non-colluding cryptographic secret providers. However, both protocols are still not scalable because each worker needs to compute a shared-secret key and a noise mask for every other client. When recovering from failures, all live clients are notified and send their masks to the server, which introduces significant communication overhead. In contrast, workers in our scheme are freed from coordinating with other workers, which leads to a more scalable system.

**Byzantine-Robust Aggregation/SGD.** Blanchard et al. (2017) first proposes Krum and Multi-Krum for training machine learning models in the presence of Byzantine workers. Mhamdi et al. (2018) proposes a general enhancement recipe termed *Bulyan*. Alistarh et al. (2018) proves a robust SGD training scheme with optimal sample complexity and the number of SGD computations. Muñoz-González et al. (2019) uses HMM to detect and exclude Byzantine workers for federated learning. Yin et al. (2018) proposes median and trimmed-mean based robust algorithms which achieve optimal statistical performance. For robust learning on non-i.i.d dataset only appear recently (Li et al., 2019; Ghosh et al., 2019; He et al., 2020). Further, Xie et al. (2018) generalizes the Byzantine attacks to manipulate data transfer between workers and server and Xie et al. (2019a) extends it to tolerate an arbitrary number of Byzantine workers.

Pillutla et al. (2019) proposes a robust aggregation rule RFA which is also privacy preserving. However, it is only robust to data poisoning attack as it requires workers to compute aggregation weights according to the protocol. Corrigan-Gibbs & Boneh (2017) proposes a private and robust aggregation system based on secret-shared non-interactive proof (SNIP). Despite the similarities between our setups, the generation of a SNIP proof on client is expansive and grows with the dimensions. Besides, this paper offers limited robustness as it only validates the range of the data.

**Inference As A Service.** An orthogonal line of work is inference as a service or oblivious inference. A user encrypts its own data and uploads it to the server for inference. (Gilad-Bachrach et al., 2016; Rouhani et al., 2017; Hesamifard et al., 2017; Liu et al., 2017; Mohassel & Zhang, 2017; Chou et al., 2018; Juvekar et al., 2018; Riazi et al., 2019) falls into a general category of 2-party computation (2PC). A number of issues have to be taken into account: the non-linear activations should be replaced with MPC-friendly activations, represent the floating number as integers. Ryffel et al. (2019) uses functional encryption on polynomial networks. Gilad-Bachrach et al. (2016) also have to adapt activations to polynomial activations and max pooling to scaled mean pooling.

**Server-Aided MPC.** One common setting for training machine learning model with MPC is the server-aided case (Mohassel & Zhang, 2017; Chen et al., 2019). In previous works, both the model weights and the data are stored in shared values, which in turn makes the inference process computationally very costly. Another issue is that only a limited number of operations (function evaluations) are supported by shared values. Therefore, approximating non-linear activation functions again introduces significant overhead. In our paper, the computation of gradients are local to the

workers, only output gradients are sent to the servers. Thus no adaptations of the worker's neural network architectures for MPC are required.

## 7 CONCLUSION

In this paper, we propose a novel secure and Byzantine-robust aggregation framework. To our knowledge, this is the first work to address these two key properties jointly. Our algorithm is simple and fault tolerant and scales well with the number of workers. In addition, our framework holds for any existing distance-based robust rule. Besides, the communication overhead of our algorithm is roughly bounded by a factor of 2 and the computation overhead, as shown in Algorithm 3, is marginal and can even be computed prior to training.

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

# Appendix

## A  PROOFS

**Theorem I** (Privacy for **S1**). *Let $z = \sum_{i=1}^{n} p_i \boldsymbol{x}_i$ where $\{p_i\}_{i=1}^{n}$ is the output of byzantine oracle or a vector of 1s (non-private). Let $BV_{ij} = \langle \boldsymbol{a}_{ij}, \boldsymbol{b}_{ij}, \boldsymbol{c}_{ij} \rangle$ and $BVp_i = \langle \boldsymbol{a}_i^p, \boldsymbol{b}_i^p, \boldsymbol{c}_i^p \rangle$ be the Beaver's triple used in the multiplications. Let $\langle \cdot \rangle^{(1)}$ be the share of the secret-shared values $\langle \cdot \rangle$ on **S1**. Then for all workers $i$*

$$\mathbb{P}(\boldsymbol{x}_i = x_i \mid \{\langle \boldsymbol{x}_i \rangle^{(1)}, \langle p_i \rangle^{(1)}\}_{i=1}^{n}, \{BV_{i,j}^{(1)}, \boldsymbol{x}_i - \boldsymbol{x}_j - \boldsymbol{a}_{ij}, \boldsymbol{x}_i - \boldsymbol{x}_j - \boldsymbol{b}_{ij}\}_{i<j},$$

$$\{\langle \|\boldsymbol{x}_i - \boldsymbol{x}_j\|^2 \rangle^{(1)}\}_{i<j}, \{BVp_i^{(1)}, p_i - \boldsymbol{a}_i^p, p_i - \boldsymbol{b}_i^p\}_{i=1}^{n}, \boldsymbol{z}) = \mathbb{P}(\boldsymbol{x}_i = x_i | \boldsymbol{z})$$

*Note that the conditioned values are what **S1** observes throughout the algorithm. $\{BV_{ij}^{(1)}, \boldsymbol{x}_i - \boldsymbol{x}_j - \boldsymbol{a}_{ij}, \boldsymbol{x}_i - \boldsymbol{x}_j - \boldsymbol{b}_{ij}\}_{i<j}$ and $\{BVp_i^{(1)}, p_i - \boldsymbol{a}_i^p, p_i - \boldsymbol{b}_i^p\}_{i=1}^n$ are intermediate values during shared values multiplication.*

*Proof.* First, we use the independence of Beaver's triple to simplify the conditioned term.

- The Beaver's triples are data-independent. Since $\langle \boldsymbol{a}_i^p \rangle^{(2)}$ and $\langle \boldsymbol{b}_i^p \rangle^{(2)}$ only exist in $\{p_i - \boldsymbol{a}_i^p, p_i - \boldsymbol{b}_i^p\}_i$ and they are independent of all other variables, we can remove $\{p_i - \boldsymbol{a}_i^p, p_i - \boldsymbol{b}_i^p\}_i$ from conditioned terms.
- For the same reason $\{BVp_i^{(1)}\}_{i=1}^n$ are independent of all other variables and can be removed.
- The secret shares of aggregation weights $\langle p_i \rangle^{(1)} := (p_i + \eta_i)/2$ and $\langle p_i \rangle^{(2)} := (p_i - \eta_i)/2$ where $\eta_i$ is random noise. Then $\{\langle p_i \rangle^{(1)}\}_i$ are independent of all other variables. Thus it can be removed.

Now the left hand side (LHS) can be simplified as

$$
\begin{aligned}
LHS = &\mathbb{P}(\boldsymbol{x}_i = x_i | \{\langle \boldsymbol{x}_i \rangle^{(1)}\}_{i=1}^n, \\
&\{BV_{i,j}^{(1)}, \boldsymbol{x}_i - \boldsymbol{x}_j - \boldsymbol{a}_{ij}, \boldsymbol{x}_i - \boldsymbol{x}_j - \boldsymbol{b}_{ij}, \\
&\langle \|\boldsymbol{x}_i - \boldsymbol{x}_j\|^2 \rangle^{(1)}\}_{i<j}, \boldsymbol{z})
\end{aligned} \tag{3}
$$

There are other independence properties:

- The secret shares of the input $\langle \boldsymbol{x}_i \rangle$ can be seen as generated by random noise $\xi_i$. Thus $\langle \boldsymbol{x}_i \rangle^{(1)} := (\xi_i + \boldsymbol{x}_i)/2$ and $\langle \boldsymbol{x}_i \rangle^{(2)} := (-\xi_i + \boldsymbol{x}_i)/2$ are independent of others like $\boldsymbol{x}_i$. Besides, for all $j \neq i$, $\langle \boldsymbol{x}_i \rangle^{(\cdot)}$ and $\langle \boldsymbol{x}_j \rangle^{(\cdot)}$ are independent.
- Beaver's triple $\{BV_{i,j}^{(1)}\}_{i<j}$ and $\{BV_{i,j}^{(2)}\}_{i<j}$ are clearly independent. Since they are generated before the existance of data, they are always independent of $\{\boldsymbol{x}_j^{(\cdot)}\}_j$.

Next, according to Beaver's multiplication Algorithm 3,

$$
\langle \|\boldsymbol{x}_i - \boldsymbol{x}_j\|^2 \rangle^{(1)} = \boldsymbol{c}_{ij}^{(1)} + (\boldsymbol{x}_i - \boldsymbol{x}_j - \boldsymbol{a}_{ij})\boldsymbol{b}_{ij}^{(1)} + (\boldsymbol{x}_i - \boldsymbol{x}_j - \boldsymbol{b}_{ij})\boldsymbol{a}_{ij}^{(1)}
$$

we can remove this term from condition:

$$
\begin{aligned}
LHS = \mathbb{P}(\boldsymbol{x}_i = x_i | &\{\langle \boldsymbol{x}_i \rangle^{(1)}\}_{i=1}^n, \boldsymbol{z}, \\
&\{BV_{i,j}^{(1)}, \boldsymbol{x}_i - \boldsymbol{x}_j - \boldsymbol{a}_{ij}, \boldsymbol{x}_i - \boldsymbol{x}_j - \boldsymbol{b}_{ij}\}_{i<j})
\end{aligned} \tag{4}
$$

By the independence between $\langle \boldsymbol{x}_i \rangle^{(\cdot)}$ and $BV_{ij}^{(\cdot)}$, we can further simplify the conditioned term

$$
\begin{aligned}
LHS = \mathbb{P}(\boldsymbol{x}_i = x_i | &\{\langle \boldsymbol{x}_i \rangle^{(1)}\}_{i=1}^n, \boldsymbol{z}, \\
&\{BV_{i,j}^{(1)}, \langle \boldsymbol{x}_i - \boldsymbol{x}_j - \boldsymbol{a}_{ij} \rangle^{(2)}, \langle \boldsymbol{x}_i - \boldsymbol{x}_j - \boldsymbol{b}_{ij} \rangle^{(2)}\}_{i<j})
\end{aligned} \tag{5}
$$

Since $BV_{ij}^{(1)}$ and $BV_{ij}^{(2)}$ are always independent of all other variables, we know that

$$
LHS = \mathbb{P}(\boldsymbol{x}_i = x_i | \{\langle \boldsymbol{x}_i \rangle^{(1)}\}_{i=1}^n, \boldsymbol{z}) \tag{6}
$$

For worker $i$, $\forall j \neq i$, $\langle \boldsymbol{x}_i \rangle^{(\cdot)}$ and $\langle \boldsymbol{x}_j \rangle^{(1)}$ are independent

$$
LHS = \mathbb{P}(\boldsymbol{x}_i = x_i | \boldsymbol{z}). \qquad \square
$$

**Theorem II** (Privacy for S2). *Let $\{p_i\}_{i=1}^n$ is the output of byzantine oracle or a vector of 1s (non-private). Let $BV_{ij} = \langle \boldsymbol{a}_{ij}, \boldsymbol{b}_{ij}, \boldsymbol{c}_{ij} \rangle$ and $BVp_i = \langle \boldsymbol{a}_i^p, \boldsymbol{b}_i^p, \boldsymbol{c}_i^p \rangle$ be the Beaver's triple used in the multiplications. Let $\langle \cdot \rangle^{(2)}$ be the share of the secret-shared values $\langle \cdot \rangle$ on **S2**. Then for all workers $i$*

$$
\begin{aligned}
&\mathbb{P}(\boldsymbol{x}_i = x_i \mid \{\langle \boldsymbol{x}_i \rangle^{(2)}, \langle p_i \rangle^{(2)}, p_i\}_{i=1}^n, \{BV_{i,j}^{(2)}, \boldsymbol{x}_i - \boldsymbol{x}_j - \boldsymbol{a}_{ij}, \boldsymbol{x}_i - \boldsymbol{x}_j - \boldsymbol{b}_{ij}\}_{i<j}, \\
&\{\langle \|\boldsymbol{x}_i - \boldsymbol{x}_j\|^2 \rangle^{(2)}, \|\boldsymbol{x}_i - \boldsymbol{x}_j\|^2\}_{i<j}, \{BVp_i^{(2)}, p_i - \boldsymbol{a}_i^p, p_i - \boldsymbol{b}_i^p\}_{i=1}^n) \\
&= \mathbb{P}(\boldsymbol{x}_i = x_i \mid \{\|\boldsymbol{x}_i - \boldsymbol{x}_j\|^2\}_{i<j})
\end{aligned} \tag{1}
$$

*Note that the conditioned values are what **S2** observed throughout the algorithm. $\{BV_{ij}^{(2)}, \boldsymbol{x}_i - \boldsymbol{x}_j - \boldsymbol{a}_{ij}, \boldsymbol{x}_i - \boldsymbol{x}_j - \boldsymbol{b}_{ij}\}_{i<j}$ and $\{BVp_i^{(2)}, p_i - \boldsymbol{a}_i^p, p_i - \boldsymbol{b}_i^p\}_{i=1}^n$ are intermediate values during shared values multiplication.*

*Proof.* Similar to the proof of Theorem I, we can first conclude

- $\{p_i - \boldsymbol{a}_i^p, p_i - \boldsymbol{b}_i^p\}_i$ and $\{BV p_i^{(2)}\}_{i=1}^n$ could be dropped because these they are data independent and no other terms depend on them.
- $\{\langle p_i \rangle^{(2)}\}_{i=1}^n$ is independent of the others so it can be dropped.
- $\{p_i\}_{i=1}^n$ can be inferred from $\{\|\boldsymbol{x}_i - \boldsymbol{x}_j\|^2\}_{ij}$ so it can also be dropped.
- By the definition of $\{\langle \|\boldsymbol{x}_i - \boldsymbol{x}_j\|^2 \rangle^{(2)}\}_{ij}$, it can be represented by $\{\boldsymbol{x}_i\}^{(2)}$ and $\{BV_{ij}^{(2)}, \boldsymbol{x}_i - \boldsymbol{x}_j - \boldsymbol{a}_{ij}, \boldsymbol{x}_i - \boldsymbol{x}_j - \boldsymbol{b}_{ij}\}_{i<j}$.

Now the left hand side (LHS) can be simplified as

$$
\begin{aligned}
LHS =& \mathbb{P}(\boldsymbol{x}_i = x_i | \{\langle \boldsymbol{x}_i \rangle^{(2)}\}_{i=1}^n, \\
& \{BV_{ij}^{(2)}, \boldsymbol{x}_i - \boldsymbol{x}_j - \boldsymbol{a}_{ij}, \boldsymbol{x}_i - \boldsymbol{x}_j - \boldsymbol{b}_{ij}, \\
& \|\boldsymbol{x}_i - \boldsymbol{x}_j\|^2\}_{i<j})
\end{aligned}
\tag{7}
$$

Because $\boldsymbol{x}_i$ is independent of $\{\langle \boldsymbol{x}_i \rangle^{(2)}\}_{i=1}^n$ as well as data independent terms like $\{BV_{ij}^{(2)}, \boldsymbol{a}_{ij}^{(1)}, \boldsymbol{b}_{ij}^{(1)}\}_{i<j}$, we have

$$
LHS = \mathbb{P}(\boldsymbol{x}_i = x_i \,|\, \|\boldsymbol{x}_i - \boldsymbol{x}_j\|^2\}_{i<j}) \qquad \square
$$

**Theorem III** (from DP to LDP). *Suppose that the noise $\nu_t$ in (2) is sufficient to ensure that the set of model parameters $\{\boldsymbol{w}_t\}_{t \in [T]}$ satisfy $(\varepsilon, \delta)$-DP for $\varepsilon \geq 1$. Then, running (2) with using Alg. 1 to compute $(\boldsymbol{x}_t + \eta_t)$ by securely aggregating $\{\boldsymbol{x}_{1,t} + n\eta_t, \boldsymbol{x}_{2,t}, \ldots, \boldsymbol{x}_{n,t}\}$ satisfies $(\varepsilon, \delta)$-LDP.*

*Proof.* Suppose that worker $i \in [n]$ copmutes it gradient $\boldsymbol{x}_i$ based on data $d_i \in \mathcal{D}$. For the sake of simplicity, let us assume that the arregate model satisfies $\varepsilon$-DP. The proof is identical for the more relaxed notion of $(\varepsilon, \delta)$-DP fo r$\varepsilon \geq 1$. This implies that for any $j \in [n]$ and $d_j, \tilde{d}_j \in \mathcal{D}$,

$$
\frac{\Pr\left[ \frac{1}{n}(\sum_{i=1}^n \boldsymbol{x}_i(d_i)) + \nu = \boldsymbol{y} \right]}{\Pr\left[ \frac{1}{n}(\sum_{i \neq j} \boldsymbol{x}_i(d_i)) + \frac{1}{n}\boldsymbol{x}_j(\tilde{d}_j) + \nu = \boldsymbol{y} \right]} \leq \varepsilon, \forall \boldsymbol{y}.
\tag{8}
$$

Now, we examine the communication received by each server and measure how much information is revealed about any given worker $j \in [n]$. The values stored and seen are:

- **S1**: The secret share $(\boldsymbol{x}_1 + n\nu)^{(1)}$, $\{\boldsymbol{x}_i(d_i)^{(1)}\}_{i=2}^n$ and the sum of other shares $(\boldsymbol{x}_1 + n\nu)^{(2)} + \sum_{i=2}^n \boldsymbol{x}_i(d_i)^{(2)} = ((\sum_{i=1}^n \boldsymbol{x}_i(d_i)) + n\nu)^{(2)}$.
- **S2**: The secret share $(\boldsymbol{x}_1 + n\nu)^{(2)}$, $\{\boldsymbol{x}_i(d_i)^{(2)}\}_{i=2}^n$.
- Worker $i$: $\boldsymbol{z} = (\sum_{i=1}^n \boldsymbol{x}_i(d_i)) + n\nu$.

The equality above is because our secret shares are *linear*. Now, the values seen by any worker satisfy $\varepsilon$-LDP directly by (8). For the server, note that by the definition of our secret shares, we have for any worker $j$,

$$
\boldsymbol{x}_j(d_j)^{(1)} \text{ is independent of } \boldsymbol{x}_j(d_j)
$$
$$
\Rightarrow \Pr[\boldsymbol{x}_j(d_j)^{(1)} = y] = \Pr[\boldsymbol{x}_j(d_j)^{(1)} = \tilde{y}], \forall \boldsymbol{y}, \tilde{\boldsymbol{y}}
$$
$$
\Rightarrow \Pr[\boldsymbol{x}_j(d_j)^{(1)} = y] = \Pr[\boldsymbol{x}_j(\tilde{d}_j)^{(1)} = y], \forall d_j, \tilde{d}_j \in \mathcal{D}.
$$

A similar statement holds for the second share. This proves that the values computed/seen by the workers or servers satisfy $\varepsilon$-LDP. $\qquad \square$

## B   NOTES ON SECURITY

### B.1   BEAVER'S MPC PROTOCOL

In this section, we briefly introduce Beaver (1991)'s classic implementations of addition $\langle x + y \rangle$ and multplication $\langle xy \rangle$ given additive secret-shared values $\langle x \rangle$ and $\langle y \rangle$ where each party $i$ holding $x_i$ and $y_i$. The algorithm for multiplication is given in Algorithm 3.

---

**Algorithm 3** Beaver (1991)'s MPC Protocol

---

**input**: $\langle x \rangle$; $\langle y \rangle$; Beaver's triple $(\langle a \rangle, \langle b \rangle, \langle c \rangle)$ s.t. $c = ab$
**output**: $\langle z \rangle$ s.t. $z = xy$
**for all** party $i$ **do**
    locally compute $x_i - a_i$ and $y_i - b_i$ and then broadcast them to all parties
    collect all shares and reveal $x - a = \Sigma_i(x_i - a_i)$, $y - b = \Sigma_i(y_i - b_i)$
    compute $z_i := c_i + (x - a)b_i + (y - b)a_i$
**end for**
The first party 1 updates $z_1 := z_1 + (x - a)(y - b)$

---

*Addition.* The secret-shared values form of sum, $\langle x + y \rangle$, is obtained by simply each party $i$ locally compute $x_i + y_i$.

*Multiplication.* Assume we already have three secret-shared values called a triple, $\langle a \rangle$, $\langle b \rangle$, and $\langle c \rangle$ such that $c = ab$.

Then note that if each party broadcasts $x_i - a_i$ and $y_i - b_i$, then each party $i$ can compute $x - a$ and $y - b$ (so these values are publicly known), and hence compute

$$z_i := c_i + (x - a)b_i + (y - b)a_i$$

Additionally, one party (chosen arbitrarily) adds on the public value $(x - a)(y - b)$ to their share so that summing all the shares up, the parties get

$$\Sigma_i z_i = c + (x - a)b + (y - b)a + (x - a)(y - b) = xy$$

and so they have a secret sharing $\langle z \rangle$ of $xy$.

**The generation of Beaver's triples.** There are many different implementations of the offline phase of the MPC multiplication. For example, semi-homomorphic encryption based implementations (Keller et al., 2018) or oblivious transfer-based implementations (Keller et al., 2016). Since their security and performance have been demonstrated, we may assume the Beaver's triples are ready for use at the initial step of our protocol.

### B.2 Notes on obtaining a secret share

Suppose that we want to secret share a bounded real vector $\boldsymbol{x} \in (-B, B]^d$ for some $B \geq 0$. Then, we sample a random vector $\xi$ uniformly from $(-B, B]^d$. This is easily done by sampling each coordinate independently from $(-B, B]$. Then the secret shares become $(\xi, \boldsymbol{x} - \xi)$. Since $\xi$ is drawn from a uniform distribution from $[-B, B]^d$, the distribution of $x - \xi$ conditioned on $\boldsymbol{x}$ is still uniform over $(-B, B]^d$ and (importantly) independent of $\boldsymbol{x}$. All arithmetic operations are then carried out modulo $[-B, B]$ i.e. $B + 1 \equiv -B + 1$ and $-B - 1 \equiv B - 1$. This simple scheme ensures information theoretic input-privacy for continuous vectors.

The scheme described above requires access to true randomness i.e. the ability to sample uniformly from $(-B, B]$. We make this assumption to simplify the proofs and the presentation. We note that differential privacy techniques such as (Abadi et al., 2016) also assume access to a similar source of true randomness. In practice, however, this would be replaced with a pseudo-random-generator (PRG) (Blum & Micali, 1984; Yao, 1982).

### B.3 Computational indistinguishability

Let $\{X_n\}$, $\{Y_n\}$ be sequences of distributions indexed by a security parameter $n$ (like the length of the input). $\{X_n\}$ and $\{Y_n\}$ are *computationally indistinguishable* if for every polynomial-time A and polynomially-bounded $\varepsilon$, and sufficiently large $n$

$$\left| \Pr[A(X_n) = 1] - \Pr[A(Y_n) = 1] \right| \leq \varepsilon(n) \tag{9}$$

If a pseudorandom generator, instead of true randomness, is used in Appendix B.2 , then the shares are indistinguishable from a uniform distribution over a field of same length. Thus in Theorem I and Theorem II, the secret shares can be replaced by an independent random variable of uniform distribution with negligible change in probability.

### B.4    NOTES ON THE SECURITY OF **S2**

Theorem II proves that **S2** does not learn anything besides the pairwise distances between the various models. While this does leak some information about the models, **S2** cannot use this information to reconstruct any $x_i$. This is because the pair-wise distances are invariant to translations, rotations, and shuffling of the coordinates of $\{x_i\}$.

This remains true even if **S2** additionally learns the global model too.

## C    DATA OWNERSHIP DIAGRAM

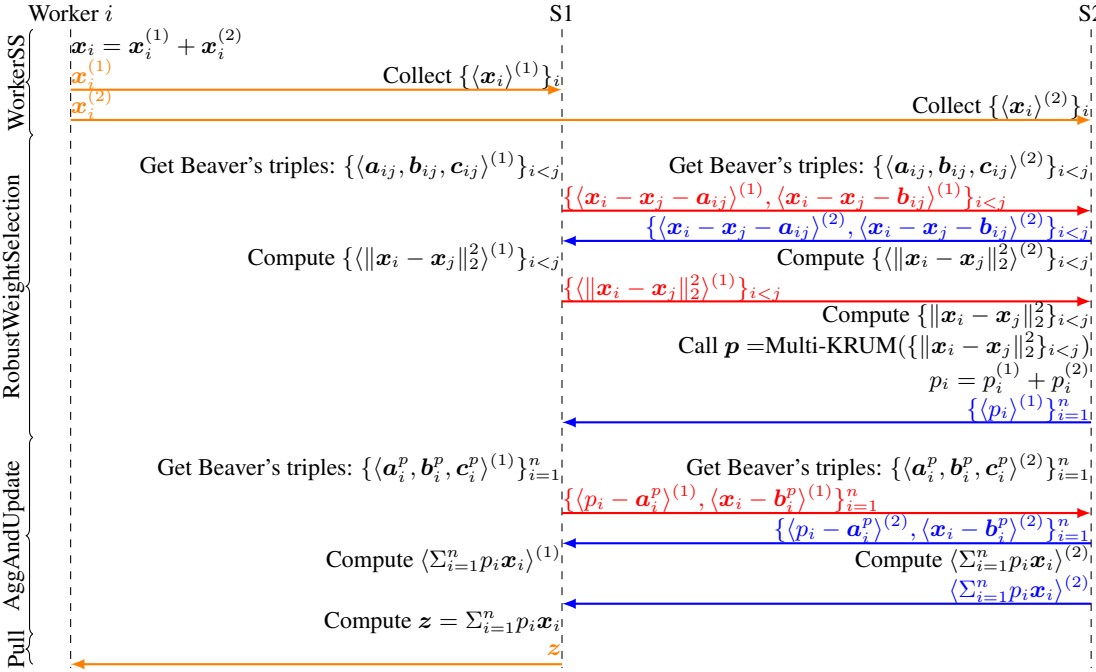

Figure 3: Overview of data ownership and Algorithm 1. The underlying Byzantine-robust oracle is Multi-Krum.

In Figure 3, we show a diagram of data ownership to demonstrate of the data transmitted among workers and servers. Note that the Beaver's triples are already local to each server so that no extra communication is needed.

# D  THREE SERVER MODEL

In this section, we introduce a robust algorithm with information-theoretical privacy guarantee at the cost of more communication between servers. We avoid exposing pairwise distances to **S2** by adding to the system an additional non-colluding server, the **crypto provider**(Wagh et al., 2019). A crypto provider does not receive shares of gradients, but only assists other servers for the multiparty computation. Now our pipeline for one aggregation becomes: 1) the workers secret share their gradients into 2 parts; 2) the workers send their shares to **S1** and **S2** respectively; 3) **S1**, **S2** and the crypto provider compute the robust aggregation rule using crypto primitives; 4) servers reveal the output of aggregation and send back to workers.

The (Wagh et al., 2019) use crypto provider to construct efficient protocols for the training and inference of neural networks. In their setup, workers secret share their samples to the servers and then servers secure compute a neural network. In contrast, we consider the federated learning setup where workers compute the gradients and servers perform a multiparty computation of a (robust) aggregation function. The aggregated neural network is public to all. As the (robust) aggregation function is much simpler than a neural netwuork, our setup is more computationally efficient. Note that we can directly plug our secure robust aggregation rule into their pipeline and ensure both robustness and privacy-preserving in their setting.

The crypto provider enables servers to compute a variety of functions on secret shared values.

- MATMUL: Given $\langle \boldsymbol{x} \rangle$ and $\langle \boldsymbol{y} \rangle$, return $\langle \boldsymbol{x}^\top \boldsymbol{y} \rangle$. The crypto provider generates and distribute beaver's triple for multiplication.
- PRIVATECOMPARE: Given $\langle x \rangle$ and a number $r$, reveal a bit $(x > r)$ to **S1** and **S2**, see Algorithm 5. This can be directly used to compare $\langle x \rangle$ and $\langle y \rangle$ by comparing $\langle x - y \rangle$ and 0.
- SELECTSHARE: Given $\langle \boldsymbol{x} \rangle$ and $\langle \boldsymbol{y} \rangle$ and $\alpha \in \{0, 1\}$, return $\langle (1-\alpha)\boldsymbol{x} + \alpha \boldsymbol{y} \rangle$, see Algorithm 6. This function can be easily extended to select one from more quantities.

The combination of PRIVATECOMPARE and SELECTSHARE enables sorting scalar numbers, like distances. Thus we can use these primitives to compute Krum on secret-shared values. For other aggregation rules like RFAPillutla et al. (2019), we need other primitives like division. We refer to Wagh et al. (2019) for more primitives like division, max pooling, ReLU. We also leave the details of the three aforementioned primitives in Appendix D.1.

**Three server MultiKrum.** In Algorithm 4 we present a three-server MULTIKRUM algorithm. First, **S1**, **S2**, and **S3** compute the pairwise distances $\{\langle d_{ij} \rangle\}_{ij}$, but do not reveal it like Algorithm 2. For each $i$, we use PRIVATECOMPARE and SELECTSHARE to sort $\{\langle d_{ij} \rangle\}_j$ by their magnitude. Then we compute $\langle \text{score}_i \rangle$ using SELECTSHARE. Similarly we sort $\{\langle \text{score}_i \rangle\}_i$ and get a selection vector $\boldsymbol{\alpha}$ for workers with lowest scores. Finally, we open $\langle \boldsymbol{\alpha} \cdot \mathbf{X} \rangle$ and reveal $\sum_{i \in \mathcal{I}} \boldsymbol{\alpha}_i \boldsymbol{x}_i$ to everyone.

We remark that sorting the $\{\langle d_{ij} \rangle\}_j$ does not leak anything about their absolute or relative magnitude. This is because: 1) **S3** picks $i, j$ from $\pi_1, \pi_2$ which is unknown to **S1** and **S2**; 2) **S3** encodes $i, j$ into a selection vector $\boldsymbol{\alpha}_{ij}$ and secret shares it to **S1** and **S2**; 3) For **S1** and **S2**, they only observe secret-shared selection vectors which is computationally indistinguishable from a random string. Thus **S1** and **S2** learn nothing more than the outcome of MultiKrum. On the other hand, PRIVATECOMPARE guarantees the crypto provider **S3** does not know the results of comparison. So **S3** also knows nothing more than the output of MultiKrum. Thus Algorithm 4 enjoys information-theoretical security.

## D.1  THREE SERVER MODEL IN SECURENN

**Changes in the notations.** The Algorithm 6 and Algorithm 5 from SecureNN use different notations. For example they use $\langle w \rangle_j^p$ to represent the share $j$ of $w$ in a ring of $\mathbb{Z}^p$. Morever, the Algorithm 5 secret shares each bit of a number $x$ of length $\ell$ which writes $\{\langle x[i] \rangle^p\}_{i \in [\ell]}$. The $\oplus$ means xor sum.

---

**Algorithm 4** Three Server MULTIKRUM

---

**Input:** S1 and S2 hold $\{\langle \boldsymbol{x}_i \rangle^{(0)}\}_i$ and $\{\langle \boldsymbol{x}_i \rangle^{(1)}\}_i$ resp. $f, m$
**Output:** $\sum_{i \in \mathcal{I}} \boldsymbol{\alpha}_i \boldsymbol{x}_i$ where $\mathcal{I}$ is the set selected by MULTIKRUM

**On S1 and S2 and S3:**
**For** $i$ in $1 \ldots n$ **do**
    **For** $j \neq i$ in $1 \ldots n$ **do**
        Compute $\langle \boldsymbol{x}_i - \boldsymbol{x}_j \rangle$ locally on S1 and S2
        Call $\mathcal{F}_{\text{MATMUL}}(\{S1, S2\}, S3)$ with $(\langle \boldsymbol{x}_i - \boldsymbol{x}_j \rangle, \langle \boldsymbol{x}_i - \boldsymbol{x}_j \rangle)$ and get $\langle d_{ij} \rangle = \langle \|\boldsymbol{x}_i - \boldsymbol{x}_j\|^2 \rangle$
    **End for**
**End for**
Denote $\mathbf{d} = [d_{ij}]_{i<j}$ be a vector of the distances and $\mathbf{X} = [\boldsymbol{x}_1; \ldots; \boldsymbol{x}_n]$

**On S3:**
Let $\pi_1$ and $\pi_2$ be 2 random permutation function.
**For** $i$ in $\pi_1(1 \ldots n)$ **do**
    **For** $j \neq i$ in $\pi_2(1 \ldots n)$ **do**
        Let $\boldsymbol{\alpha}_{ij}$ be the selection vector of $\mathbf{d}$ whose entry for $d_{ij}$ is 1 and all others are 0.
        Compute $\langle \boldsymbol{\alpha}_{ij} \rangle$ and send $\langle \boldsymbol{\alpha}_{ij} \rangle_0$ to S1 and send $\langle \boldsymbol{\alpha}_{ij} \rangle_1$ to S2.
        Call Algorithm 6 with input $(\langle \boldsymbol{\alpha}_{ij} \rangle, \{\langle d_{ij} \rangle\}_{i<j})$ and get $\langle d'_{ij} \rangle$. $(d_{ij} = \langle d'_{ij} \rangle^{(0)} + \langle d'_{ij} \rangle^{(1)})$
    **End for**
    Sort $\{\langle d'_{ij} \rangle\}_j$ using Algorithm 5 to compute $\langle \text{score}_i \rangle = \sum_{i \to j} \langle d'_{ij} \rangle$
**End for**
Sort $\{\langle \text{score}_i \rangle\}_i$ using Algorithm 5 and record the $m$ indicies $\mathcal{I}$ with lowset scores.
Let $\boldsymbol{\alpha}$ be a selection vector of length $n$ so that for all entry $i \in \mathcal{I}$ are 1 and all others are 0.
Compute $\langle \boldsymbol{\alpha} \rangle$ and send $\langle \boldsymbol{\alpha} \rangle^{(0)}$ to S1 and send $\langle \boldsymbol{\alpha} \rangle^{(1)}$ to S2.
Compute $\langle \boldsymbol{\alpha} \cdot \mathbf{X} \rangle$ using $\mathcal{F}_{\text{MATMUL}}$.

**On S1 and S2:**
Let $k = 0$ for S1 and $k = 1$ for S2
**For** $\tilde{i}$ in $1 \ldots n$ **do**
    **For** $\tilde{j}$ in $1 \ldots (n-1)$ **do**
        Receive $\langle \boldsymbol{\alpha}_{**} \rangle^{(k)}$ from S3
        Call Algorithm 6 with input $(\langle \boldsymbol{\alpha}_{**} \rangle, \{\langle d_{ij} \rangle\}_{i<j})$ and get $\langle d'_{*\tilde{j}} \rangle$.
    **End for**
    Sort $\{\langle d'_{*\tilde{j}} \rangle\}_{\tilde{j}}$ using Algorithm 5 to compute $\langle \text{score}_{\tilde{i}} \rangle = \sum_{\tilde{i} \to \tilde{j}} \langle d'_{*\tilde{j}} \rangle$
**End for**
Sort $\{\langle \text{score}_{\tilde{i}} \rangle\}_{\tilde{i}}$ using Algorithm 5.
Receive $\langle \boldsymbol{\alpha} \rangle_k$
Compute $\langle \boldsymbol{\alpha} \cdot \mathbf{X} \rangle$ using $\mathcal{F}_{\text{MATMUL}}$.
**S1 and S2:** Open $\langle \boldsymbol{\alpha} \cdot \mathbf{X} \rangle$ to reveal $\sum_{i \in \mathcal{I}} \boldsymbol{\alpha}_i \boldsymbol{x}_i$

---

---

**Algorithm 5** PRIVATECOMPARE $\Pi_{\text{PC}}(\{S_1, S_2\}, S_3)$ (Wagh et al., 2019, Algo. 3)

---

**Input:** $S_1$ and $S_2$ hold $\{\langle x[i]\rangle_0^p\}_{i \in [\ell]}$ and $\{\langle x[i]\rangle_1^p\}_{i \in [\ell]}$, respectively, a common input $r$ (an $l$ bit integer) and a common random bit $\beta$. The superscript $p$ is a small prime number like 67.
**Output:** $S_3$ gets a bit $\beta \oplus (x > r)$
**Common Randomness:** $S_1, S_2$ hold $\ell$ common random value $s_i \in \mathbb{Z}_p^*$ for all $i \in [\ell]$ and a random permutation $\pi$ for $\ell$ elements. $S_1$ and $S_2$ additionally hold $\ell$ common random values $u_i \in \mathbb{Z}_p^*$.

**On each** $j \in \{0, 1\}$ **server** $S_{j+1}$

---
Let $t = r + 1 \mod 2^\ell$
**for** $i = \{\ell, \ell-1, \ldots, 1\}$ **do**
    **if** $\beta = 0$ **then**
        $\langle w_i \rangle_j^p = \langle x[i] \rangle_j^p + jr[i] - 2r[i]\langle x[i] \rangle_j^p$
        $\langle c_i \rangle_j^p = jr[i] - \langle x[i] \rangle_j^p + j + \sum_{k=i+1}^{\ell} \langle w_k \rangle_j^p$
    **else if** $\beta = 1$ **AND** $r \neq 2^\ell - 1$ **then**
        $\langle w_i \rangle_j^p = \langle x[i] \rangle_j^p + jt[i] - 2t[i]\langle x[i] \rangle_j^p$
        $\langle c_i \rangle_j^p = -jt[i] + \langle x[i] \rangle_j^p + 1 - j + \sum_{k=i+1}^{\ell} \langle w_k \rangle_j^p$
    **else**
        If $i \neq 1$, $\langle c_i \rangle_j^p = (1-j)(u_i + 1) - ju_i$, else $\langle c_i \rangle_j^p = (-1)^j \cdot u_i$.
    **end if**
**end for**
Send $\{\langle d_i \rangle_j^p\}_i = \pi(\{s_i \langle c_i \rangle_j^p\}_i)$ to $S_3$

**On server $S_3$**

---
For all $i \in [\ell]$, $S_3$ computes $d_i = \textbf{Reconst}^p(\langle d_i \rangle_0^p, \langle d_i \rangle_1^p)$ and sets $\beta' = 1$ iff $\exists i \in [\ell]$ such that $d_i = 0$
$S_3$ outputs $\beta'$

---

**Algorithm 6** SELECTSHARE $\Pi_{\text{SS}}(\{S_1, S_2\}, S_3)$ (Wagh et al., 2019, Algo. 2)

---

**Input:** $S_1$ and $S_2$ hold $(\langle \alpha \rangle_0^L, \langle x \rangle_0^L, \langle y \rangle_0^L)$ and $(\langle \alpha \rangle_1^L, \langle x \rangle_1^L, \langle y \rangle_1^L)$, resp.
**Output:** $S_1$ and $S_2$ get $\langle z \rangle_0^L$ and $\langle z \rangle_1^L$ resp., where $z = (1 - \alpha)x + \alpha y$.
**Common Randomness:** $S_1, S_2$ hold shares of 0 over $\mathbb{Z}_L$ denoted by $u_0$ and $u_1$.
For $j \in \{0, 1\}$, $S_{j+1}$ compute $\langle w \rangle_j^L = \langle y \rangle_j^L - \langle x \rangle_j^L$.
$S_1, S_2, S_3$ call $\mathcal{F}_{\text{MATMUL}}(\{S_1, S_2\}, S_3)$ with $S_{j+1}, j \in \{0, 1\}$ having input $(\langle \alpha \rangle_j^L, \langle w \rangle_j^L)$ and $S_1$, $S_2$ learn $\langle c \rangle_0^L$ and $\langle c \rangle_1^L$, resp.
For $j \in \{0, 1\}$, $S_{j+1}$ outputs $\langle z \rangle_j^L = \langle x \rangle_j^L + \langle c \rangle_j^L + u_j$

---

# E EXAMPLE: TWO-SERVER PROTOCOL WITH BYZANTINESGD ORACLE

We can replace MultiKrum with ByzantineSGD in (Alistarh et al., 2018). To fit into our protocol, we make some minor modifications but still guarantee that output is same. The core part of (Alistarh et al., 2018) is listed in Algorithm 7.

---

**Algorithm 7** ByzantineSGD (Alistarh et al., 2018)

---

**input**: $\mathcal{I}$ is the set of good workers, $\{A_i\}_{i\in[m]}$, $\{\|B_i - B_j\|\}_{i<j}$ $\{\|\nabla_{k,i} - \nabla_{k,j}\|\}_{i<j}$ $(i, j \in [m])$, thresholds $\mathfrak{T}_A, \mathfrak{T}_B > 0$
**output**: Subset good workers $\mathcal{S}$
$A_{\text{med}} := \text{median}\{A_1, \ldots, A_m\}$;
$B_{\text{med}} \leftarrow B_i$ where $i \in [m]$ is any machine s.t. $|\{j \in [m] : \|B_j - B_i\| \le \mathfrak{T}_B\}| > m/2$;
$\nabla_{\text{med}} \leftarrow \nabla_{k,i}$ where $i \in [m]$ is any machine s.t. $|\{j \in [m] : \|\nabla_{k,j} - \nabla_{k,i}\| \le 2\nu\}| > m/2$;
$\mathcal{S} \leftarrow \{i \in \mathcal{I} : |A_i - A_{\text{med}}| \le \mathfrak{T}_A \wedge \|B_i - B_{\text{med}}\| \le \mathfrak{T}_B \wedge \|\nabla_{k,j} - \nabla_{k,i}\| \le 4\nu\}$;

---

The main algorithm can be summarized in Algorithm 8, the red lines highlights the changes. Different from Multi-Krum (Blanchard et al., 2017), Alistarh et al. (2018) uses states in their algorithm. As a result, the servers need to keep track of such states.

---

**Algorithm 8** Two-Server Secure ByzantineSGD

---

**Setup**:
- $n$ workers, at most $\alpha$ percent of which are Byzantine.
- Two non-colluding servers **S1** and **S2**
- ByzantineSGD Oracle: returns an indices set $\mathcal{S}$.
    - With thresholds $\mathfrak{T}_A$ and $\mathfrak{T}_B$
    - Oracle state $A_i^{\text{old}}, \langle B_i^{\text{old}}\rangle$ for each worker $i$

**Workers**:
1. (**WorkerSecretSharing**):
    (a) randomly split private $\boldsymbol{x}_i$ into additive secret shares $\langle \boldsymbol{x}_i\rangle = \{\boldsymbol{x}_i^{(1)}, \boldsymbol{x}_i^{(2)}\}$ (such that $\boldsymbol{x}_i = \boldsymbol{x}_i^{(1)} + \boldsymbol{x}_i^{(2)}$)
    (b) sends $\boldsymbol{x}_i^{(1)}$ to **S1** and $\boldsymbol{x}_i^{(2)}$ to **S2**

**Servers**:
1. $\forall i$, **S1** collects gradient $\boldsymbol{x}_i^{(1)}$ and **S2** collects $\boldsymbol{x}_i^{(2)}$.
    (a) Use Beaver's triple to compute $A_i := \langle\langle\boldsymbol{x}_i\rangle, \langle\boldsymbol{w} - \boldsymbol{w}_0\rangle\rangle_{\text{inner}} + A_i^{\text{old}}$
    (b) $\langle B_i\rangle := \langle\boldsymbol{x}_i\rangle + \langle B_i^{\text{old}}\rangle$
2. (**RobustSubsetSelection**):
    (a) For each pair $(i, j)$ of gradients computes their distance $(i < j)$:
        - On **S1** and **S2**, compute $\langle B_i - B_j\rangle = \langle B_i\rangle - \langle B_j\rangle$ locally
        - Use precomputed Beaver's triple and Algorithm 3 to compute the distance $\|B_i - B_j\|^2$
        - On **S1** and **S2**, compute $\langle\boldsymbol{x}_i - \boldsymbol{x}_j\rangle = \langle\boldsymbol{x}_i\rangle - \langle\boldsymbol{x}_j\rangle$ locally
        - Use precomputed Beaver's triple and Algorithm 3 to compute the distance $\|\boldsymbol{x}_i - \boldsymbol{x}_j\|_2^2$
    (b) **S2** perform Byzantine SGD $\mathcal{S}$=ByzantineSGD($\{A_i\}_i, \{\|B_i - B_j\|\}_{i<j}, \{\|\boldsymbol{x}_i - \boldsymbol{x}_j\|\}_{i<j}, \mathfrak{T}_A, \mathfrak{T}_B$); if $|\mathcal{S}| < 2$, exit; Convert $\mathcal{S}$ to a weight vector $\boldsymbol{p}$ of length $n$
    (c) **S2** secret-shares $\langle\boldsymbol{p}\rangle$ with **S1**
3. (**AggregationAndUpdate**):
    (a) On **S1** and **S2**, use MPC multiplication to compute $\langle\sum_{i=1}^n p_i\boldsymbol{x}_i\rangle$ locally
    (b) **S2** sends its share of $\langle\sum_{i=1}^n p_i\boldsymbol{x}_i\rangle^{(2)}$ to **S1**
    (c) **S1** reveals $\boldsymbol{z} = \sum_{i=1}^n p_i\boldsymbol{x}_i$ to all workers.
    (d) **S2** updates $A_i^{\text{old}} \leftarrow A_i, \langle B_i^{\text{old}}\rangle \leftarrow \langle B_i\rangle$

**Workers**:
1. (**WorkerPullModel**): Collect $\boldsymbol{z}$ and update model $\boldsymbol{w} \leftarrow \boldsymbol{w} + \boldsymbol{z}$ locally

---

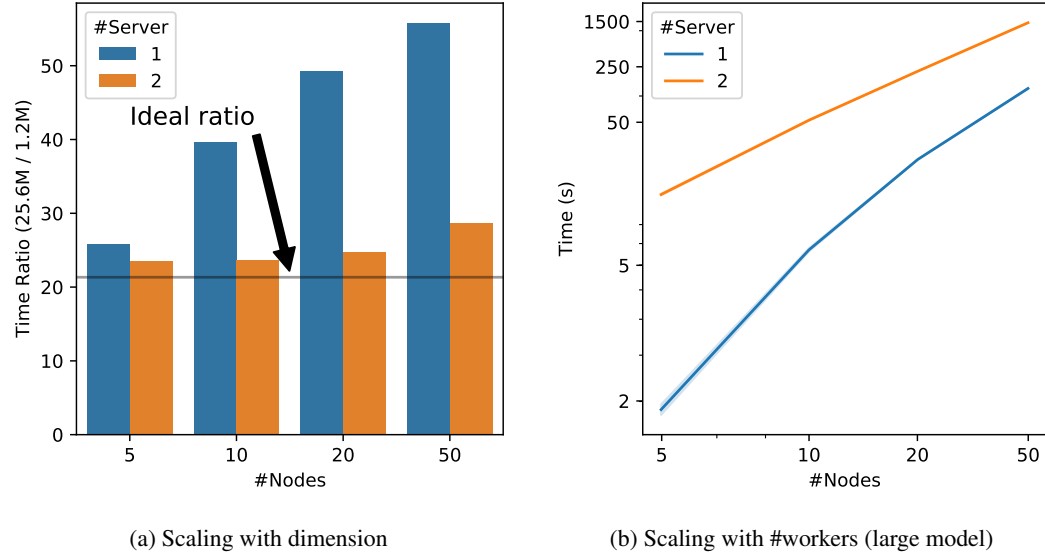

(a) Scaling with dimension           (b) Scaling with #workers (large model)

Figure 4: Scaling two-server model and one-server model to 5, 10, 20, 50 nodes.

## F   ADDITIONAL EXPERIMENTS

We benchmark the performance of our two-server protocol with one-server protocol on the google kubernetes engine. We create a cluster of 8 nodes (machine-type=e2-standard-2) where 2 servers are deployed on different nodes and the workers are deployed evenly onto the rest 6 nodes. We run the experiments with 5, 10, 20, 50 workers and a large model of 25.6 million parameters (similar to ResNet-56) and a small model of 1.2 million parameters. We only record the time spent on communication and aggregation (krum). We benchmark each experiment for three times and take their average. The results are shown in Figure 4.

**Scaling with dimensions.** In Figure 4a, we compute the ratio of time spent on large model and small model. We can see that the ratio of two-server model is very close to the ideal ratio which suggests it scales linearly with dimensions. This is expected because krum scales linearly with dimension. For aggregation rules based on high-dimensional robust mean estimation, we can remove the dependence on $d$. We leave it as a future work to incorporate more efficient robust aggregation functions.

**Scaling with number of workers.** In Figure 4b, we can see that the time spent on both one-server and two-server model grow with $O(n^2)$. However, we notated that this complexity comes from the aggregation rule we use, which is krum, not from our core protocol. For other aggregation rules like ByzantineSGD Alistarh et al. (2018), the complexity of aggregation rule is $O(n)$ and we can observe better scaling effects. We leave it as a future work to incorporate and benchmark more efficient robust aggregation rules.

**Setups.** Note that in our experiments, the worker-to-server communication and server-to-server communication has same bandwidth of 1Gb/s. In the realistic application, the link between servers can be infiniband and the bandwidth between worker and server are typically smaller. Thus, this protocol will be more efficient than we have observed here.

