# OpenReview forum: "Secure Byzantine-Robust Machine Learning"
_ICLR.cc/2021/Conference — Reject_

### Official Review · AnonReviewer4 · 2020-10-28
**Failed assumptions regarding secure computation**

**Rating:** 3
**Confidence:** 5

**Review:**

Summary:
The paper presents a system for collaborative training where two central servers compute the model update using two-party computation. There are two variants, one where the update are not checked, and one where the server use algorithm (Aggr) to weight the updates such that outliers are excluded.

Pros:
The basic ideas are presented succinctly, and the overall setup solves a relevant problem.

Cons:
- The main issue I have is that the authors claim in Section 4.3 that they "can" use full-precision (real) values in the computation but the underlying techniques (secret sharing and Beaver triples) have only been proposed for integer/quantized values. In particular, if the actual value is small, adding a large random value will override it due to rounding in floating-point representation. I cannot find any treatment of this issue or any reference to works to have tackled floating-point secure computation such as Aliasgari et al., NDSS '13.
- Similarly, the claim that the paper does not rely on cryptographic primitives seems exaggerated given the use of secret sharing and Beaver triples.
- The cost of generating Beaver triples is ignored. The paper does not estimate how many are needed.
- Claiming Byzantine robustness seems too strong when the two central servers have to be semi-honest.

Conclusion:
I recommend rejection because of the irrealistic approach regarding real-valued computation. The authors should either present credible secure floating-point computation or change their claim to quantized computation.

Minor issues:
3.1: x_i +- \xi_i is not an additive secret sharing of x_i but 2*x_i.
3.1: sum of (the) other share
3.1: "sum of other share" should be \sum *p_i* x_i^(2)?
3.1: It is not surprising that S2 does not learn anything in secure computation.
3.3: In particular, (unfinished sentence)
3.3: don't face
4.2: vetor (twice)
6: median and trimmed-mean -based (odd positioning of dash)

---

> ### Author Response · Authors · 2020-11-15
> **Assuming full precision is standard in analyses of ML methods**
>
> We thank the reviewer for their close reading of our work. The major concern seems to be the realism of the schemes proposed. We admittedly work with a simplified model, where we have access to full precision numbers (as opposed to floating point). However, this is par for the course in optimization and deep learning and we don’t think this diminishes our contribution. We expand upon the individual points below and believe that we address all the concerns raised. We request the reviewer to re-evaluates our work in light of our replies below and also with a broader ML (and not just cryptography/security) community in mind.
>
> > Analysis of security in floating-point of secret sharing and Beaver triplets schemes.
>
> Firstly, our schemes are fully realizable and secure assuming access to full precision computation. We give simple schemes for both Beaver triplets (Section B.1) and secret sharing schemes (Section B.2). Note that the secret sharing scheme in B.2 assumes that all vectors have a known bounded norm. This is also a prevalent assumption in differential privacy and is commonly enforced simply by clipping the gradient (Abadi et al., 2016). In contrast, schemes which rely on homomorphic encryption, or Yao’s garbled circuits are inherently designed for integers/quantized values unlike ours.
>
> Secondly, assuming access to full precision is standard in ML, mathematical optimization, and differential privacy (Abadi et al., 2016). This is because, unlike in traditional cryptographic applications, deep learning methods are very robust to rounding errors (Gupta et al., 2015 [1]) and additional noise (Neelakantan et al., 2016 [2]), with current models routinely trained in 16-bit precision [3]. Thus while rounding errors may occur when using our schemes in floating-point, these do not affect the convergence of our schemes.
>
> Finally, we agree that we do not investigate additional leakage of information due to the use of floating-point, nor do we provide a secure floating-point implementation, but have added your suggested reference (Aliasgari et al. 2013) for this case. We have written a new remark in Section 3.4 and hope that our work will inspire more follow up analysis from security experts.
>
> > Reliance on and cost of cryptographic schemes.
>
> We wanted to convey that our schemes only use simple arithmetic operations (addition and multiplication) as opposed to e.g. zero-knowledge proofs, key agreement, homomorphic encryption, or garbled circuits. We have changed our claim to state we do not use *heavy* cryptographic primitives.
>
> While we do require Beaver’s triplets to be computed, these are independent of the data and can be precomputed during an *offline* stage by the two servers. The number of triplets required is O(dn^2) for Krum and O(dn) for RFA. Our scheme is extremely computationally light from the perspective of the workers, who are typically much more resource constrained, as e.g. in federated learning. Further, optimizing the generation of Beaver’s triples has itself been a well-studied topic with established benchmarks (see for e.g. in Table 2 of (Mohassel & Zhang, 2017), or (Blanchard et al., 2017)).
>
> > Cannot claim Byzantine robustness with semi-honest servers.
>
> Byzantine robustness is always with respect to the workers. All prior works in the area use the notion of Byzantine in this same sense (Blanchard et al., 2017; Yin et al., 2018; Pillutla et al., 2019;...). Even when dealing with privacy and security for training and federated learning, the assumption of semi-honest servers is standard (Bonawitz et al., 2017; Mohassel & Zhang, 2017;...), and of significant interest to the entire ICLR community.
>
> In summary, it has been commonly held that (quoting Bagdasaryan et al., 2020) *“Robust aggregation mechanisms ... are incompatible with secure aggregation.”* We challenge this belief and make significant progress combining the two.
>
> Additional References:
>
> [1] Gupta, S., Agrawal, A., Gopalakrishnan, K., & Narayanan, P. Deep learning with limited numerical precision. ICML 2015.
>
> [2] Neelakantan, A., Vilnis, L., Le, Q. V., Kaiser, L., Kurach, K., Sutskever, I., & Martens, J.. Adding Gradient Noise Improves Learning for Very Deep Networks. ICLR 2016.
>
> [3] https://www.tensorflow.org/guide/mixed_precision

---

### Official Review · AnonReviewer2 · 2020-10-28
**A good paper that tackles both privacy and adversarial machines**

**Rating:** 7
**Confidence:** 3

**Review:**

**Paper summary**

1. The paper introduces a two server protocol to handle privacy concerns and Byzantine threats in a Federated Learning system simultaneously.
2. In the protocol, each client secretly shares their model update with the two servers by splitting its model update such that neither server can know what the model update is without colluding with the other server. The servers are able to compute pairwise distances of all updates securely. These distances are used by byzantine robust aggregators to find a robust model update.


**Strengths**
1. The paper handles two very relevant and important issues with Federated Learning simultaneously - privacy and Byzantine resilience (which includes data poisoning attacks).
2. The proposed algorithm is shown to have theoretical guarantees.
3. A wide array of byzantine robust aggregation rules can be incorporated easily into the framework.
5. There isn't much communication overhead between the workers and the server.
6. The two server protocol does not seem to be too difficult to implement in practice. The algorithm requires that the two servers should not collude. This requirement does not seem too difficult to be enforced onto big companies that will use FL.
7. In the absence of Byzantine machines, the non-robust protocol seems to be compatible with differential privacy.

**Concerns**
1. Can it be quantified (using some information theoretic bound) that only pairwise distances cannot leak much information? In the worst case, it can leak some information. For example if the pairwise distance is 0 for some pair, then each machine exactly knows the other's model update. However, I believe that using some assumption on distribution of model updates, we can still get some information theoretic guarantee. The three server algorithm in Appendix D seems to address this, but I wonder if we can say something for the two server algorithm too.
2. Can it be extended to dimension-independent robust mean estimation techniques? The paper only considers distance based aggregators (with the exception of  (Alistarh et al., 2018), which I discuss later). These aggregators all seem to suffer from an error that grows with dimensions as $\sqrt{d}$ (see section 2 in (Wang et al., 2020). A recent line of works has given robust mean estimators that have errors that are dimension independent (Dong et al., 2019; Diakonikolas et al., 2017; Diakonikolas et al., 2016; Lai et al., 2016). However these use second order information like the empirical covariance matrix too. Can the algorithm proposed in this paper be extended to these algorithms too?

Alistarh et al. (2018) also give dimension independent guarantees, but they require the workers to sample a new data point at every iteration, which may not hold for many FL systems.

**Score justification**

The paper tackles a very relevant problem for Federated Learning. Further, the proposed algorithm has nice theoretical guarantees and it looks like the algorithm can be easily implemented in a variety of large FL systems.


**References**

Wang, L., Pang, Q., Wang, S. and Song, D., 2020. F2ED-Learning: Good Fences Make Good Neighbors. arXiv preprint arXiv:2010.01175.

Dong, Y., Hopkins, S. and Li, J., 2019. Quantum entropy scoring for fast robust mean estimation and improved outlier detection. In Advances in Neural Information Processing Systems (pp. 6067-6077).

Diakonikolas, I., Kamath, G., Kane, D.M., Li, J., Moitra, A. and Stewart, A., 2017. Being robust (in high dimensions) can be practical. arXiv preprint arXiv:1703.00893.

Diakonikolas, I., Kamath, G., Kane, D.M., Li, J., Moitra, A. and Stewart, A., 2016, October. Robust Estimators in High Dimensions without the Computational Intractability. In 2016 IEEE 57th Annual Symposium on Foundations of Computer Science (FOCS) (pp. 655-664).

Lai, K.A., Rao, A.B. and Vempala, S., 2016, October. Agnostic estimation of mean and covariance. In 2016 IEEE 57th Annual Symposium on Foundations of Computer Science (FOCS) (pp. 665-674). IEEE.

---

> ### Author Response · Authors · 2020-11-15
> **Reply R2**
>
> We thank Reviewer 2 for reading our work and insightful opinions. We address the concerns as follows:
>
> - As is mentioned by the reviewer, the best way to address the pairwise distance leakage is to use the protocol in Appendix D which achieves information-theoretic guarantees. For the two-server case, quantifying the exact privacy leakage is hard. But it is reasonable to believe that the adversaries can not reconstruct the other worker’s inputs. Note that this information is leaked only to server 2, which does not have access to any other information. Thus even if distance=0, the server can figure out that two workers have the same update, but cannot figure out what the actual update was. The workers do not gain any information whatsoever.
>
> - It is very interesting to combine security with dimension-independent robust aggregation rules. However, this is quite challenging because of the difficulty of the secure computation of eigenvalue/eigenvectors. A possible direction could be to implement a secure version of the power method, but we leave this for future work.

---

### Official Review · AnonReviewer1 · 2020-10-29
**Novel proposal of robust and secure federated learning**

**Rating:** 5
**Confidence:** 2

**Review:**

This work proposes a method to robustly (<.5 adversarial workers) aggregate model updates using two non-colluding servers. The proposed method scales well with the number of workers and is compatible with local DP and different robust aggregation protocols. Especially the scalability is a big improvement compared to previous methods. The authors discuss related work that relies on public key infrastructure and requires pairwise secrets between clients. One big advantage of the proposed protocol is that there is no communication between the workers.

Positives
- The paper is very well written and easy to follow.
- The contributions are significant in terms of scalability, compared to other protocols discussed in the literature review.
- The theoretical justifications seem sound and are relatively easy to follow, although I didn't verify the proofs in detail.

Negatives
- The assumptions are still strong (non-colluding servers), but I am not very familiar with the literature to fully assess the implications. It would be great to put the strong assumption of non-colluding servers more into context. How does it compare to other protocols? (somewhat discussed in the lit section)
- The experimental section is lacking a bit. It's not entirely clear what additional insights the provided analysis gives, especially also given the unrealistic setup of only five workers. It's not clear what the actual experiments are, referred to with S1 Avg, S2 Avg, S2 Krum (S1/S2 also refer to the servers in the rest of the paper). Is S2 Avg the proposed protocol, just w/o the robust computation? A more detailed discussion why the robust version increases the server communication significantly (even though the authors claim it doesn't. AFAICT it almost doubles the total cost) would be welcome. Is it because of the Beaver's triple communication to share the pairwise distances? (TBH, I am not sure this paper really needed an experimental section, but the current one has serious gaps).

While the paper proposes a novel idea with significant improvements, the paper is lacking wrt putting it into context of the existing field and a more detailed interpretation of the experiments would be welcome. For these reasons the recommendation is to reject the paper in it's current state.

---

> ### Author Response · Authors · 2020-11-15
> **Reply R1**
>
> We thank the reviewer for the suggestions, which we are glad to incorporate. We will first give more context of our scheme, and then address issues individually. We will address the reviewer’s concerns as follows:
>
> - While we agree that assuming that the servers do not collude is not ideal, this is a very standard and common setting in privacy-preserving machine learning (Mohassel et al., 2017; Wagh et al., 2018; Corrigan-Gibbs et al., 2017; Wagh et al., 2020 [1]). Practically, non-collusion can be ensured by “government regulations or other social deterrents which are sufficient enforcers” (Wagh et al., 2020 [1]). This model is also seeing real-world implementations e.g. a secret-sharing based multi-party setup similar to ours has been adopted in the sugar beet auction in Denmark (Bogetoft et al., 2009 [2]).
>
>   Alternative crypto primitives (oblivious transfer, garbled circuits) are used for two-party computation while distributed training is typically multi-party computation. Therefore the secret-sharing is more widely used for distributed machine learning. On the other hand, using secret-sharing with one server means asking workers to compute pairwise distances and send them to the server. However, it has serious problems comparing to two-server:
>   1. They have significantly higher communication overhead because worker to worker communication is much slower than the server to server.
>   2. Dealing with fault tolerance and straggler is more complicated and time-consuming (Bonawitz et al., 2017).
>   3. One server scheme can be not compatible with robust aggregation rules (Bagdasaryan et al., 2020).
>
>   Overall, the two-server model is efficient and its non-colluding assumptions are not strong which makes it suitable for robust aggregation.
>
> - We have also considered not adding an experiment section. Since our scheme guarantees to give the same convergence rate/outcome as the non-secure version, we ignore the epoch-to-accuracy curve. Benchmarking secret sharing based algorithms, on the other hand, has already been conducted in (Mohassel et al., 2017) and (Wagh et al., 2020 [1]), and the results suggest secret-sharing is much more efficient than its alternatives. The number of triples required (thus the number of communication) is O(dn^2) for the Krum aggregation rule (Blanchard et al., 2017), and O(dn) for the more efficient RFA aggregation rule (Pillutla et al., 2019).
>
>   In the end, we chose to put a simulation in this paper to demonstrate that the overhead of our primitive is small. This is in contrast to other crypto primitives like zero-knowledge proof which takes 0.03 second to process a vector of 100 integers (Corrigan-Gibbs et al., 2017).
>
>   We agree with the points made by the reviewer about the experiments and we are runing a new experiment on Google cloud with a larger cluster. We don’t have good competing methods yet for the aforementioned reasons, so the goal of the new experiments is to set a baseline for future work. We will update this thread as soon as the results are ready.
>
> Additional References:
>
> [1]: Wagh, Sameer, et al. "FALCON: Honest-Majority Maliciously Secure Framework for Private Deep Learning." arXiv preprint arXiv:2004.02229 (2020).
>
> [2]: Bogetoft, Peter, et al. "Secure multiparty computation goes live." International Conference on Financial Cryptography and Data Security. Springer, Berlin, Heidelberg, 2009.
>
> ================================================================================
>
> We have updated the experiments in Appendix F.

---

### Official Review · AnonReviewer3 · 2020-10-29
**When all assumptions hold a nice method**

**Rating:** 6
**Confidence:** 4

**Review:**



The paper proposes a method combining privacy and byzantine-robustness for distnace-based aggregation.  This is a relevant and interesting topic.

The paper assumes that it is not a problem that one of the server learns all pairwise distances.  Still, every known pairwise distance |&x_i-x_j|| more tightly connects the relative position of the points x_i together.  If the dimension of the vectors x_i isn't large, then knowledge of a limited set of x_i vectors may allow this server to deduce all other vectors (or may allow him to determine significant constraints all other vectors x_j must satisfy).

Once we accept the above assumption, the paper seems mostly sound, sometimes with minor issues in writing or precision (a few examples below).
The approach doesn't introduce fundamentally new techniques but combines existing ideas to realize a more powerful solution.  The text is quite well written.


DETAILS

* "WorkerSecretSharing (Figure 1a): ... This can be done e.g. by sampling a large noise \xi_i and then using x_i \pm \xi_i as the shares." -> as x_i must be the sum of the shares, I guess you mean the shares are (x_i+\xi_i)/2 and (x_i-\xi_i)/2
* Section 3.2: RobustWeightSelection (Figure 1b): seleced subset -> selected subset
* Section 3.2: RobustWeightSelection (Figure 1b): "S2 secret shares with S1 the values of {<p_i>}" -> please make clear what exactly is "secret-shares", it is not the secret sharing used by workers to split x=x_1+x_2.   (I guess you describe here step 3b in algo 2)
* "Compatibility with local differential privacy. One byproduct of our protocol can be used to convert differentially private mechanisms, such as (Abadi et al., 2016) which only of the aggregate model which guarantees privacy," -> the last "which only ..." subphrase needs a verb

---

> ### Author Response · Authors · 2020-11-15
> **Reply R3**
>
> We thank the reviewer for reading our work and the suggestions. We are glad to incorporate the suggestions of the reviewer. We will address the issues as follows:
>
> - Yes. We mean the shares are  $(x_i+\xi_i)/2$ and $(x_i-\xi_i)/2$. We will correct that.
> - Thanks for pointing out the typo.
> - By saying "S2 secret shares with S1 the values of {<p_i>}", we mean S2 splits each p_i into two parts $(p_i + \xi_i)/2$ and $(p_i - \xi_i)/2$ where the $\xi$ is different and independent from the worker randomness. Then S2 sends a share, e.g.  $(p_i + \xi_i)/2$, to S1 such that S1 does not know $p_i$ (Step 2c of Algorithm 2). Finally, both S1 and S2 are holding shares of $\{p_i\}$ and $\{x_i\}$, so that they can securely compute weighted average $\sum_i p_i x_i$  through multiplication (using Beaver’s triple) (Step 3a of Algorithm 2). We polish the description of algorithm to make it more clear.
> - Thanks for the proofreading.

---

### Decision · Program_Chairs · 2021-01-07
**Final Decision**

**Decision:**

Reject

**Comment:**

This paper presents a secure aggregation method to ensure byzantine robustness. The reviewers thought that the idea was interesting, but had the following concerns.
* Relaxing the assumptions used in the theoretical analysis as much as possible
* Run more extensive experiments
I encourage the authors to their feedback into account when preparing the revised draft.